# DNA-Based Electrodes and Computational Approaches on the Intercalation Study of Antitumoral Drugs

**DOI:** 10.3390/molecules26247623

**Published:** 2021-12-16

**Authors:** Edson Silvio Batista Rodrigues, Isaac Yves Lopes de Macêdo, Giovanna Nascimento de Mello e Silva, Arthur de Carvalho e Silva, Henric Pietro Vicente Gil, Bruno Junior Neves, Eric de Souza Gil

**Affiliations:** 1Lafam—Laboratory for Pharmaceutical and Environmental Analysis, Faculdade de Farmácia, Universidade Federal de Goiás, Goiânia 74605-170, Brazil; edson.silvio.b@gmail.com (E.S.B.R.); isaacyvesl@gmail.com (I.Y.L.d.M.); giovannamellonutri@gmail.com (G.N.d.M.e.S.); 2LabMol—Laboratory for Molecular Modeling and Drug Design, Faculdade de Farmácia, Universidade Federal de Goiás, Goiânia 74605-170, Brazil; arthurcs.farma@gmail.com (A.d.C.e.S.); henricgil@discente.ufg.br (H.P.V.G.); brunoneves@ufg.br (B.J.N.)

**Keywords:** voltammetry, DNA intercalation, electrocatalysis, molecular docking, intermolecular interactions

## Abstract

The binding between anticancer drugs and double-stranded DNA (dsDNA) is a key issue to understand their mechanism of action, and many chemical methods have been explored on this task. Molecular docking techniques successfully predict the affinity of small molecules into the DNA binding sites. In turn, various DNA-targeted drugs are electroactive; in this regard, their electrochemical behavior may change according to the nature and strength of interaction with DNA. A carbon paste electrode (CPE) modified with calf thymus ds-DNA (CPDE) and computational methods were used to evaluate the drug–DNA intercalation of doxorubicin (DOX), daunorubicin (DAU), idarubicin (IDA), dacarbazine (DAR), mitoxantrone (MIT), and methotrexate (MTX), aiming to evaluate eventual correlations. CPE and CPDE were immersed in pH 7 0.1 mM solutions of each drug with different incubation times. As expected, the CPDE response for all DNA-targeted drugs was higher than that of CPE, evidencing the drug–DNA interaction. A peak current increase of up to 10-fold was observed; the lowest increase was seen for MTX, and the highest increase for MIT. Although this increase in the sensitivity is certainly tied to preconcentration effects of DNA, the data did not agree entirely with docking studies, evidencing the participation of other factors, such as viscosity, interfacial electrostatic interactions, and coefficient of diffusion.

## 1. Introduction

Intercalating drugs are classical antineoplastic agents that have DNA as the target binding site, and the interaction strength dictates the anticancer efficacy [1,2,3,4,5,6]. Among the main intercalating agents, doxorubicin (DOX), daunorubicin (DAU), idarubicin (IDA), dacarbazine (DAR), mitoxantrone (MIT), and methotrexate (MTX) (Figure 1) have been widely used in cancer therapy [2,4,5,7,8].

Despite the fact that the intercalation with DNA can lead to inhibition of mRNA transcription and duplication, many intercalating drugs act by dual or multiple mechanisms. For instance, DOX, DAU, and IDA act by inhibiting topoisomerase II, leading to the formation of double-stranded breaks of DNA [7,8,9,10,11,12,13,14,15]. On the other hand, DAR is a prodrug that needs to be metabolized to a methyldiazonium ion, which is responsible for alkylation of DNA [9,13,14]. In turn, MTX, a broad-spectrum anticancer drug, binds to and inhibits the enzyme dihydrofolate reductase, resulting in inhibition of purine nucleotide and thymidylate synthesis and, subsequently, inhibition of DNA and RNA syntheses. Though MTX is mostly classified as an antifolate agent, electrochemical assays have showed its potential for DNA intercalation [4,16].

The mechanism of interaction of chemical compounds with DNA, such as that of antidepressants [17], antibiotics [18], and especially anticancer drugs [3,4,19,20,21,22,23,24], has been intensively investigated in recent years. These interactions can be covalent or noncovalent. The noncovalent interactions can involve electrostatic forces, nonpolar intercalation, and steric adjustment [13].

Therefore, the proposal of methods to evaluate the DNA intercalation may bring insights for the development of novel antineoplastics. Owing to the electroactive nature of DNA bases (guanine and adenine) and intercalating drugs, electrochemical methods are interesting approaches to provide insights about the intensity of interactions [3,4,19,20,22,25].

Recent studies demonstrate that through voltammetric methods using DNA-modified electrodes, the intercalation effect can be observed through changes occurring in the intensity of the anodic peak currents of adenine and guanine, respectively [4,19,20,21,22,23,24,25].

Meanwhile, the IDA–dsDNA interaction was investigated by means of differential pulse voltammetry (DPV) using electrochemical ds-DNA biosensor and incubation solutions. The results indicated that the intercalation process changes the DNA morphological structure. Indeed, the higher the incubation time, the lower the DNA oxidation peaks. In turn, the electrochemical characterization of IDA was also performed at a glassy carbon electrode, in which the one-proton and one-electron irreversible oxidation process allowed the determination of the IDA’s diffusion coefficient, which was 7.47 × 10^−6^ cm^2^ s^−1^ in 0.1 M acetate buffer pH 4.3 [12].

The interaction of the anticancer drug dasatinib with double-stranded calf thymus DNA (ds-DNA) was investigated by evaluating the changes of anodic peak currents at the pencil graphite electrode (PGE) in the absence and presence of DNA. The voltammetric data allowed to obtain the kinetic parameters, i.e., the diffusion coefficient (D) and heterogeneous rate constants (ks), as well as the thermodynamic parameters [26].

Similar studies were performed for MTX, and besides kinetic parameters, it was evidenced that the planar heterocyclic ring of anthraquinone moiety and the two aminoethylamino side-chains led to higher binding constant values of this intercalator [27]. The interaction of DAU and dsDNA, as well as the effect of an inclusion complex of cyclodextrins was investigated by UV–VIS spectroscopy and square wave voltammetry. The formation of strong inclusion complexes promoting the drug–DNA intercalation was seen [8]. 

The organic–inorganic hybrid nanoflowers (NF) produced from cysteine and glutamic amino acids and Cu_3_(PO_4_)_2_ were used to modify PGE. The modified NF-PGE exhibited high catalytic properties, thus offering a great variety of applications, from DAU–DNA interaction study, to the quantitative determinations of DAU, DNA, and nucleotide derivatives [1].

The carbon electrodes have been the main option among the transducers in DNA-based biosensors. Graphene oxide (GO) was used for the DNA biosensor optimization for detection of MTX. The sensibility gain can be attributed to the richer oxygen-containing functional groups and nanostructure of GO. In turn, the adsorptive behavior of MTX at such surfaces may be higher [28]. 

A CPE modified with ds-DNA, zeolitic imidazolate frameworks, and the ionic liquid 1-butyl-3-methylimidazolium methanesulfonate was used to evaluate the interaction of the MIT with guanine bases of ds-DNA. Under optimal experimental conditions (incubation time = 12 min; pH = 4.8; acetate buffer solution and DNA = 50 mg/L), the proposed biosensor presented a detection limit of 3.0 nM. It was observed that above ambient temperature of 25 °C, ds-DNA was released from electrode surface, which was related to the decrease of binders viscosity in the carbon paste matrix. Moreover, the docking investigation of MIT into the ds-DNA sequence was carried out to propose the intercalative binding mode of the drug into the nitrogenous-based pairs of ds-DNA [29]. 

Hence, voltammetric assays with carbon paste electrode (CPE) and carbon paste electrode modified with double-stranded DNA (CPDE) can be used as fast, simple, and low-cost alternatives to monitor drug–DNA interactions. In addition, computational chemistry approaches are also extremely useful in the studies involving interactions between drugs and macromolecules. One of such approaches is the molecular docking, in which association forces and binding energies are measured between the docked molecule and the docking macromolecule [6,13]. Although many electrochemical studies have been applied on the investigation of DNA-targeted drugs, only few works have explored the relationship between docking studies and electrochemical data. Moreover, they have been exclusively applied to one specific drug [3,4,14,15,16,17,18,19,20,21,22,23,24,25,26,27,28,29,30].

In this work, we developed a new method for evaluating the intercalation of FDA-approved antineoplastic drugs with DNA using voltammetric assays using CPE and CPDE. Results were compared to the molecular docking findings.

## 2. Results

### 2.1. CPDE for Voltammetric Evaluation of Anticancer Drugs

The use of DNA-based biosensors has been widely explored in the evaluation of DNA-targeted drugs, in which theoretical aspects (binding forces, coefficient of diffusion, intercalation mechanisms, etc.) and practical applications (quantitative determination) have been proposed. Therefore, the optimal experimental conditions for such distinct applications are well established, whereas incubation time, pH, DNA proportion, and drug concentration are some major issues [14,15,16,17,18,19,20,21,22,23,24,25,26,27,28,29,30]. A forgotten approach is the concomitant evaluation of different anticancer drugs, which evaluates how much the analytical data would match with drug potency and docking studies. In this context, due to the simple procedure steps and easy renewal, the CPE methodology is an outstanding alternative, in which many modifying trials can be performed in shorter time.

Thus, the optimized DNA-based biosensor, CPDE, was employed for the evaluation of the connection between the preconcentration effect of DNA and the proposed mechanism of action for different electroactive anticancer drugs: the anthracenedione, MIT, the anthracyclines, DOX, IDA, and DAU, representing the drugs whose action mechanism is the intercalation between DNA bases, the antimetabolite, MTX, the alkylating pro-drug with no in vitro activity, DAR, and the tricyclic antidepressant, CBP, representing the nonintercalating drugs. The chosen concentration of 1 mM was established for all target drugs, focusing the method sensitivity. The assays were performed in different incubation times (10, 30, 60, 120, and 300 s), in order to evaluate the response time at ambient temperature (25 °C) and in pH 7.0 PBS solution, both taken into account in order to be closer to the physiological conditions. 

The DP voltammogram obtained with CPE and CPDE showed one main oxidation peak for DOX, DAU, and IDA at *E*_pa_ = 0.37 V corresponding to oxidation in the quinonic portion of the anthracyclines drug class, followed by a second oxidation peak, at *E*_pa_ = 0.82 V for DAU and at *E*_pa_ = 0.68 V IDA, whereas DOX presented only a shoulder, at *E*_pa_ = 0.52 V, in accordance with literature data [1,12,14,30]. In turn, anthracenedione, MIT, as well as MTX exhibited two anodic peaks, the MIT at *E*_p1a_ = 0.32 V and *E*_p2a_ = 0.57 V, and the MTX at *E*_p1a_ = 0.71 V and E_p2a_ = 0.97 V, whereas DAR and CBP showed a single anodic peak at *E*_pa_ = 0.74 and *E*_pa_ = 0.86, respectively (Figure 2).

The study was performed in triplicates, and the average RSD value was lower than 10%. Thus, the anodic peak area observed in DPV assays (Figure 2) for CPE and CPDE after incubation for 1 min in all 1 mM drug solutions evidenced the uptake ability of DNA. In comparison to CPE, the DNA-modified CPDE presented a peak current increasing by up to 11-times; the highest for MIT, the lowest for the non-anticancer drug, but also for a trycliclic drug, CBP, in which the incremental factor was lower than 1.5. Yet, the anthracyclines, DOX, DAU and IDA, presented an average peak increment of 4.5 times, whereas that of MTX was 10 and that of DAR was 6 (Figure 2, Table 1).

The increasing peak current for CPDE in comparison to CPE was observed for all studied incubation times, which may be attributed to preconcentration effect of DNA. The higher differences are related to the higher affinity of a small molecule for DNA binding sites, but also to the lower unspecific π-stacking drug interactions with graphite layers [1,28]. Moreover, the diffusion of each small molecule through the carbon paste matrix and DNA macromolecule is a relevant factor [1,9,12,26,27,28,29,30]. Therefore, besides the peak current differences observed between CPE and CPDE, the differences observed for incubation time may bring some additional insights. The effect of incubation time was higher for MIT, followed by MTX (Table 1), with aqueous solubility of around 0.1 and 1.0 mg/mL, respectively. The highest hydro solubility and lower molecular weight of MIT may favor its diffusion through carbon paste matrix and as a consequence, may explain the higher preconcentration effect of CPDE. 

However, the aqueous solubility of the anthracyclines is higher (>1 mg/mL); the molecular weight and steric hindrance related to the planar four ring structure would hamper the diffusion, whereas, in opposite, it may favor the unspecific π-stacking interactions with the graphite layers. In turn, though DAR is also poorly soluble in water (<0.1 mg/mL), its lowest molecular weight and lower potential interaction with graphite layers may allow its diffusion in DNA, increasing the differences.

In turn, CPDE had no significant increase of the signal for CPB, in comparison to the counterpart without DNA. Indeed, CBP is the only non-anticancer drug, and accordingly to our docking studies (not shown), it has no effective intercalation with ds-DNA. Thus, it can be inferred that the increase in signal observed by the other drugs may in fact be related to the intercalation with DNA. Since the drugs are indicated to different neoplasms, and present different mechanisms of action, there was no evident correlation between the IC50 values of DNA target drugs and CPDE uptake (Table 1).

Therefore, in order to evaluate, how much the suitable fit at the DNA binding site drives the preconcentration effect of the DNA-modified electrode, the molecular docking for these different drugs was carried out.

### 2.2. Molecular Docking

Understanding DNA–drug interactions at the molecular level is a key point to understand the structural bases of pharmacological activity. Molecular docking was proposed as a complementary tool of electrochemical studies to investigate the effects of organic compounds in DNA. The molecular docking was carried out using two DNA structures: B-DNA dodecamer and DNA hexamer. B-DNA is the most common double-helical structure found in nature. Usually, part of the organic drug binds to the minor groove of B-DNA structure through hydrogen-bond interactions with adenine (A) and thymine (T) without causing large structural perturbations. part from B-DNA dockings, another set of simulations were performed using a DNA hexamer structure to investigate the intercalative binding of drugs. Intercalators introduce strong structural perturbations in DNA structure by creating an intercalation site. These ligands contain planar heterocyclic scaffolds that stack between adjacent DNA base pairs through π–π stacking interactions [31,32,33,34,35,36,37,38,39,40,41,42].

The results of various docking experiments are shown in Appendix A and Figure 3. According to scores of dockings (Appendix A), test drugs preferentially bind to the DNA hexamer (CGATCG) structure as threading intercalators, since its score exceeds one order of magnitude in relation to the B-DNA. Overall, the docking protocol proved to be very efficient to predict binding modes of DOX (Figure 3a), DAU (Figure 3b), and IDA (Figure 3c). In all cases, the anthraquinone moiety is intercalated between base pairs C (5)–G (6) and C (1)–G (2) through π–π stacking interactions. The 8-hydroxyl and 11-hydroxyl of the anthraquinone moiety also form hydrogen-bond interactions with N4 of G (2) and deoxyribose of G (6), respectively. The amino sugar is located within the minor groove, providing extra sequence specificity. Eventually, the 3’-nitrogen of the amino sugar forms a hydrogen-bond interaction with carbonyl of C (5). In addition, a hydrogen-bond interaction may be observed between hydroxyethan-1-one moiety of DOX and deoxyribose of A (3) (see Figure 3a). These conformations are in accordance with the binding mode of DNA–anthracycline structures available in the PDB [38,39].

As seen in Figure 3d, DAR forms a small number of intermolecular interactions with DNA (Figure 3e), which explains its lower affinity (GlideScore = −3.81) in comparison to the previously described anthracyclines. The main interactions pointed out by docking are π–π stacking interactions with C (5)–G (6) and C (1)–G (2) and a hydrogen-bond interaction between the amide group of DAR and deoxyribose of G (2). These results indicate that our docking protocol is robust and predictive, since DAR is a prodrug, and its cytotoxic activity is due to the generation of the metabolite methyldiazonium, which methylates DNA. Therefore, DAR is not expected to bind to DNA without metabolic bioactivation. Zhang et al. [39,40,41,42] indicate that DAR could be oxidized on a glassy carbon electrode at about 0.74 V, which could be attributed to the oxidation of the amino group of DAR to its cation radical. In view of these characteristics, we can say that docking can act in a complementary way to identify assay interference compounds [34,35] in electrochemical studies.

Docking study also suggested that MIT is an efficient threading intercalator of DNA (GlideScore = −11.88). As seen in Figure 3f, the anthraquinone moiety of MIT intercalates with DNA through π–π stacking interactions between phenyl ring and C (5)–G (6) and C (1)–G (2) base pairs. Furthermore, the hydroxyl of the anthraquinone moiety also forms a hydrogen-bond interaction with deoxyribose of G (6). Interestingly, the hydroxyethylaminoethyl-amino tails of MIT are allocated within the two grooves of DNA, providing extra sequence specificity. The tail within the minor groove is within the hydrogen-bond field of A (3) and G (2) base pairs. On the other hand, the tail bound to the major groove forms hydrogen-bond interactions with deoxyribose and the phosphate group of C (1) and G (2), respectively. These results corroborate with the study of Agarwal et al. [15], which showed through Fourier-transform infrared spectroscopy (FT-IR) and ultraviolet (UV)–VIS absorption that MTX intercalates into the double helix of DNA along with its external binding to phosphate–sugar backbone.

Finally, docking studies have also shown that MTX (Figure 3f) intercalates with DNA through π–π stacking interactions between phenyl ring and C(5)–G(6) and C(1)–G(2) base pairs. On the other hand, the pentanedioic acid moiety of the MTX also forms hydrogen-bond interactions with N4 of C(1), whereas N5 of pteridine forms two hydrogen-bond interactions with carbonyls of C(5) and T(4). These results corroborate with the study of Yang et al. [42], which showed through (UV)–VIS spectroscopy and fluorescence analysis that MTX intercalates into the double helix of DNA. In addition, a thermodynamic study developed by same authors showed that MTX promotes negative enthalpy changes (ΔH) and entropy changes (ΔS), demonstrating that this drug binds to DNA via hydrogen-bond interactions and van der Waals interactions [42]. The predicted MTX–DNA binding mode also corroborates with results published by Rafique et al. [16], which showed through FT-IR the involvement of hydrogen-bond interactions in the intercalative binding of MTX between nitrogen bases [16].

## 3. Materials and Methods

### 3.1. Reagents and Solutions

All electrolyte salts, solvents, and reagents were of analytical grade, and were used without further purification. Electrolyte solutions were prepared with double-distilled Milli-Q water (conductivity ≤ 0.1 µS cm^−1^; Millipore S. A., Molsheim, France). DOX, DAU, IDA, DAR, MIT, and MTX standards (United States Pharmacopea) were used to prepare 1 mmol L^−1^ stock solution immediately prior to the experiments. Potassium chloride, disodium hydrogen phosphate, potassium hydrogen phosphate, and sodium chloride were used to prepare a 0.1 mol L^−1^ phosphate-buffered saline (PBS) solution, pH 7.0. Furthermore, DNA stock solution was prepared by dissolving calf thymus double-strand DNA (1%, *w*/*v*) in 0.1 mol L^−1^ PBS buffer at pH = 7.0 and kept at 8 °C for 24 h with frequent stirring to ensure the formation of homogenous solution.

### 3.2. Electrochemical Assays

Voltammetric measurements were performed using a potentiostat/galvanostat PGSTAT^®^ model 204 with FRA32M module (Metrohm Autolab, Utrecht, Netherlands) integrated with NOVA 2.1^®^ software. All measurements were performed in a 1 mL one-compartment electrochemical cell coupled to a three-electrode system consisting of two carbon paste electrodes (the unmodified, named as CPE, the DNA-modified CPE, named as CPDE), a Pt wire as counter electrode, and an Ag/AgCl/KClsat reference electrode (both purchased from Lab solutions, São Paulo, Brazil).

#### 3.2.1. Carbon Paste Electrodes (CPE and CPDE)

The unmodified CPE was prepared with 100 mg of graphite powder and 30 mg of mineral oil. 

The CPDE was prepared with 100 mg of graphite powder, 50 µL of double-strand calf thymus DNA solution, and 30 mg of mineral oil.

The CPE components were rigorously mixed by spatulation, in order to get the highest and most suitable dispersion, and the electrochemical area of the carbon paste electrode was of 8 mm^2^.

#### 3.2.2. Voltammetric Evaluation of Anticancer Drug Uptake

The experimental conditions for differential pulse voltammetry (DPV) were as follows: pulse amplitude of 50 mV, pulse width of 0.5 s, and scan rate of 5 mV s^−1^. 

All voltammetric assays were performed in 0.1 mol L^−1^ phosphate-buffered solution (PBS), pH 7.0.

DP voltammograms were background-subtracted and baseline-corrected to provide better visualization. All experiments were conducted in triplicates, and data were analyzed using Origin Pro 9^®^ software (Northampton, MA, USA).

The effect of unspecific π-stacking drug interactions with graphite layers was assessed by incubating CPDE in 1 mM solutions of each drug for 2 min.

The drug intercalation magnitude study was performed by incubating CPDE 1 mM solutions of each drug for 2 min.

After incubation, the CPE and CPDE were rinsed with water to remove any excess of drug solution from electrode surface.

The washed electrodes were immersed in electrochemical cell containing 1 mL of pH 7.0 0.1 M buffer solution, and DPV was immediately performed, in order to evaluate the extension of drug accumulated on each electrode.

As a negative control, we chose a non-anticancer drug (CBP), thus we were able to assess the selectivity of the assay against the monitoring of drug–DNA intercalation.

### 3.3. Molecular Docking 

Before docking studies, a grid box was created for each DNA structure using the receptor grid generation panel of the Glide (Schrödinger, LLC, New York, NY, USA) [32]. The grid box of B-DNA dodecamer was generated with the following dimensions: 9.71 Å × 22.20 Å × 8.96 Å (*x*, *y*, and *z* axes), whereas the grid box of DNA hexamer was generated with dimensions of 3.11 Å × 13.46 Å × 28.94 Å. Finally, molecular docking calculations were carried out using Extra Precision (XP) function of Glide [33,34]. During docking, DNA structures were treated as rigid while ligands were taken as flexible.

#### 3.3.1. Ligand Preparation

Chemical structures of tested drugs were retrieved from PubChem [12] and imported into Maestro workspace v10.7 (Schrödinger, LLC, New York, NY, USA, 2016). Then, chemical structures were prepared using LigPrep (Schrödinger, LLC) at neutral pH [26]. Finally, ligand conformations were generated using ConfGen [27] with the OPLS-2005 force field [8] without any constraints. The conformer with lowest potential energy of each test compound was retained as input for docking studies.

#### 3.3.2. DNA Preparation

The 3D structures of the B-DNA dodecamer (ID: 1D64) [1] and DNA hexamer (ID: 1Z3F) [28] were retrieved from Protein Data Bank (PDB) [29] and prepared using the Protein Preparation Wizard module [30] available in Maestro suite (Schrödinger LLC) as follows: (i) bond orders and formal charges were adjusted, (ii) hydrogen atoms were added to the structures, and (iii) protonation states of polar atoms were predicted by PROPKA (Schrödinger, LLC) [31] at neutral pH.

## 4. Conclusions

The findings of voltammetry and molecular docking assays corroborated for the evaluation of the intercalation between drugs and DNA. The intercalation value measured by the current in μA is at least partially correlated with the glide score value, where the more negative the value, the greater the interaction between DNA and drug, mostly leading to an increase in the signal with the CPDE, demonstrating drug–DNA intercalation. Thus, electrochemical and molecular docking studies are versatile approaches that can be used in the search for DNA target drugs. The correlation herein observed allows the stratification between DNA target drugs with higher affinity and those with very low affinity for this binding site. However, the CPDE uptake observed for same classes of drugs is not accurate to establish any precise classification of their potency.

## Figures and Tables

**Figure 1 molecules-26-07623-f001:**
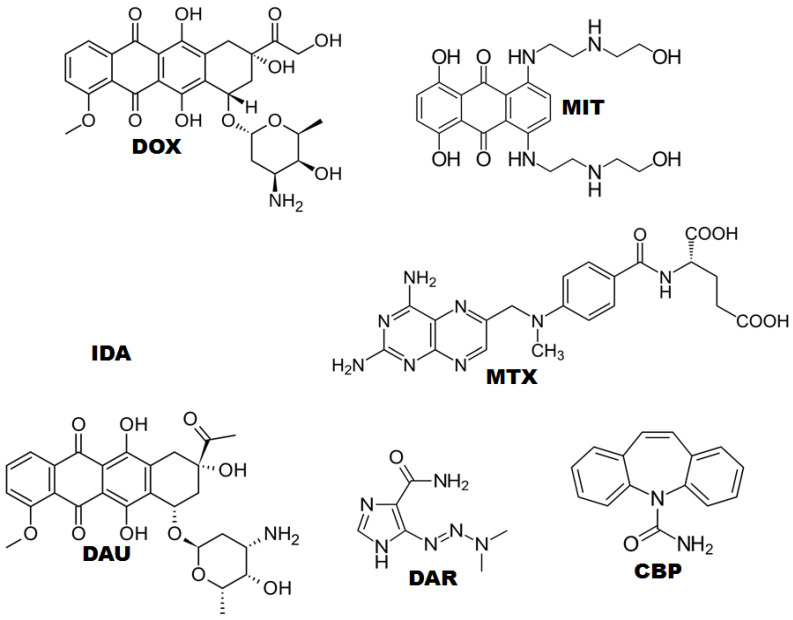
Chemical structure of anticancer drugs (DOX, IDA, DAU, MIT, MTX, and DAR) and carbamazepine (CBZ).

**Figure 2 molecules-26-07623-f002:**
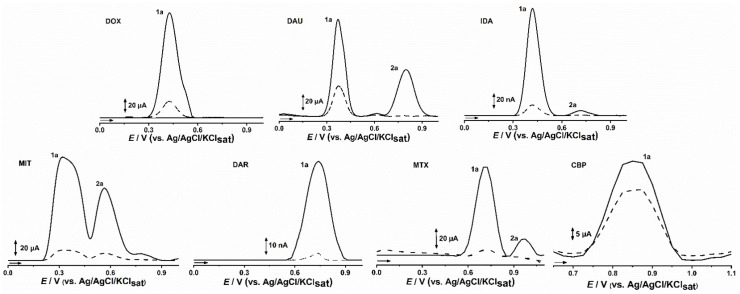
DP voltammograms obtained for CPE (– – –) CPDE (—) in pH 7.0 0.1 M PBS after 60 s immersion in 1 mM solutions of DOX, DAU, IDA, MIT, DAR, MTX, and CBP.

**Figure 3 molecules-26-07623-f003:**
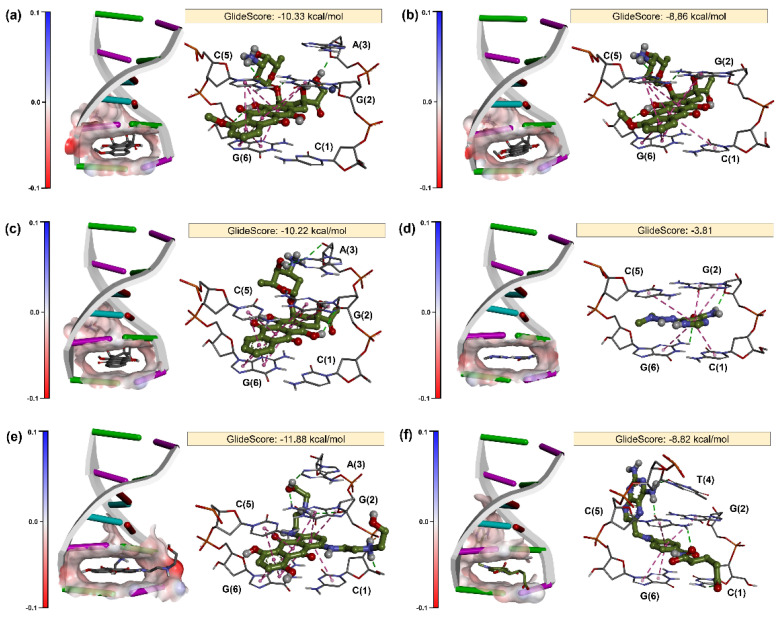
Structural basis for recognition of DOX (**a**), DAU (**b**), IDA (**c**), DAR (**d**), MIT (**e**), and MTX (**f**) by DNA hexamer (PDB ID: 1Z3F). The surface represents electron density mapping along the binding site of the DNA (CGATCG).

**Table 1 molecules-26-07623-t001:** Peak current data obtained for CPDE after different immersion times in pH 7.0 0.1 M PBS containing 1 mmol L^−1^, DOX, DAU, IDA, DAR, MIT, and MTX; current increment compared to CPE after 1 min immersion, and the corresponding IC50 values and glid scores for all drugs.

	Drugs (*I*/μA)						
Time (Seconds)	MIT	MTX	IDA	DAU	DOX	DAR	CBP
	444.481 g/mol	454.44 g/mol	497.500 g/mol	527.52 g/mol	543.52 g/mol	182.187 g/mol	236.269 g/mol
10	18	15	19	19	20	7	5
60	41	36	25	27	32	19	9
300	74	63	53	48	54	33	16
Δ*I*_t300-t10_	56	48	34	29	34	26	11
Δ*I*^CPDE-CPE^	11	10	5.0	4.7	3.8	5.6	1.4
Glid Score	−11.9	−8.9	−10.3	−8.9	−10.3	−3.8	0
IC_50 (μM)_	1.6 *	0.6 *	0.03 **	0.03 **	0.2 *	20 **	-

* https://www.cancerrxgene.org/compounds; accessed on 7 November 2021. ** https://www.selleckchem.com/products.html; accessed on 7 November 2021.

## Data Availability

Not applicable.

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
