# Peer review of "DNA-Based Electrodes and Computational Approaches on the Intercalation Study of Antitumoral Drugs"

_molecules, 2021, doi:10.3390/molecules26247623_

Round 1
Reviewer 1 Report
The manuscript (Manuscript ID: molecules-1431308) entitled with “DNA based electrodes and computational approaches on the intercalation study of antitumoral drugs” is an excellent work done by Edson Rodrigues and co-worker. But unfortunately, the manuscript is not ready for the publication at this stage because of the issues below:
# Nothing has been defined about CPE and CPDE in the Introduction, abstract as well as throughout the manuscript.
# Language is very poor and difficult to understand.
# Author has not given much attention after inserting the text to the manuscript template (there are lots of typo: such as: mitoxan-trone/ no-ticed/ in-terca-lation/ non- covalent, etc.)
# Figure 1 quality is very low.
# The format of table 1 is not correct.
Author Response
# Nothing has been defined about CPE and CPDE in the Introduction, abstract as well as throughout the manuscript.
CPE and CPDE were defined in abstract
# Language is very poor and difficult to understand. The text was entirely rewritten
# Author has not given much attention after inserting the text to the manuscript template (there are lots of typo: such as: mitoxantrone/ noticed/ intercalation/ non- covalent, etc.) We have paid more attention to this automatic DOC change
# Figure 1 quality is very low. It was improved
# The format of table 1 is not correct. It was corrected
Reviewer 2 Report
Dear Editor,
The manuscript entitled “DNA based electrodes and computational approaches on the intercalation study of antitumoral drugs” was reviewed by me. The study has quite novel, and after some revision, it can be published. For now, I cannot recommend it for publication.
Some points to be revised are listed below:
Comment
The title should be “DNA-based electrodes and computational approaches on the intercalation study of antitumoral drugs”
Abstract
Comment
To make manuscript easily readable, The full names of CPE and CPDE should be written in the abstract and then abbreviated here like “carbon paste electrode (CPE) and ……(CPDE)”. I spend much time finding the full names of CPE and CPDE.
Introduction
Comment
Page 2, second paragraph:
The interaction types of DNA and anticancer agents should be given with more detailed information. So, this paragraph should be revised.
Comment
Page 2 last paragraph:
In this paragraph, the novelty and differences of the current work should be given with detailed information and compared with the literature works.
Results and Discussion
Comment
The results in Table 1 (page 3) should be given as a diagram, not a Table, and the obtained current values must be given with the error bars that are created with a standard deviation of the measurements (the number of the measurement must be at least 3). Only one current value is not an exact result.
Comment
The lines in the caption of Figure 2 are given wrong; the lines for CPE and CPDE are the same, so one cannot understand which one CPE is or which one is CPDE.
Comment
This is only advice; I think authors could show the interaction of DNA with the drug molecule, authors could study the more concentration levels for the drugs and could show the voltammogram in Figure 2.
Comment
The conclusion part is poor, and it should be written with a more profound discussion of results.

Author Response
The title should be “DNA-based electrodes and computational approaches on the intercalation study of antitumoral drugs” We accept the good suggestion
Abstract
To make manuscript easily readable, The full names of CPE and CPDE should be written in the abstract and then abbreviated here like “carbon paste electrode (CPE) and ……(CPDE)”. I spend much time finding the full names of CPE and CPDE. CPE and CPDE were defined in abstract
Introduction
Page 2, second paragraph:
The interaction types of DNA and anticancer agents should be given with more detailed information. So, this paragraph should be revised. We included more references and improved this information
Page 2 last paragraph:
In this paragraph, the novelty and differences of the current work should be given with detailed information and compared with the literature works. Though there are many studies involving, DNA based electrodes and anticancers. There are not so many correlating electroanalysis with docking studies. Furthermore, they are at the most performed with few drugs. Our work, aim to show how much the drug affinity to DNA binding site by evaluating different drugs, correlates with the DNA uptake (pre concentration effect) observed for DNA modified electrodes
Results and Discussion
The results in Table 1 (page 3) should be given as a diagram, not a Table, and the obtained current values must be given with the error bars that are created with a standard deviation of the measurements (the number of the measurement must be at least 3). Only one current value is not an exact result. We agree, the study was performed in triplicates the average RSD value was lower than 10%.
The lines in the caption of Figure 2 are given wrong; the lines for CPE and CPDE are the same, so one cannot understand which one CPE is or which one is CPDE. I was corrected
This is only advice; I think authors could show the interaction of DNA with the drug molecule, We have performed each single molecule, and the targets were chosen taking into account different anticancer mechanisms. We also, include a tryciclic, whose pharmacologic action site is not DNA, but have some structural similarities with intercalators.
The conclusion part is poor, and it should be written with a more profound discussion of results. In fact, the correlations were not well demostrated, but we have improved
Reviewer 3 Report
The manuscript describes the intercalation study of antitumoral drugs using electrochemical method and docking tool. The novelty of this method is not suitable for the general level of Molecules because some studies were done for investigation of these drugs and DNA previously as below:
doi: 10.1002/elan.201700615
10.3389/fchem.2020.00814
10.1016/j.jpha.2016.07.005
10.1016/j.snb.2020.129120
10.1016/j.bioelechem.2014.06.002
10.1016/j.apsusc.2008.06.132
10.1016/j.bpc.2005.04.009
Author Response
The manuscript describes the intercalation study of antitumoral drugs using electrochemical method and docking tool. The novelty of this method is not suitable for the general level of Molecules because some studies were done for investigation of these drugs and DNA previously as below:
doi: 10.1002/elan.201700615 A Graphene Oxide-DNA Electrochemical Sensor Based on Glassy Carbon Electrode for Sensitive Determination of Methotrexate
10.3389/fchem.2020.00814 A DNA Based Biosensor Amplified With ZIF-8/Ionic Liquid Composite for Determination of Mitoxantrone Anticancer Drug: An Experimental/Docking Investigation
10.1016/j.jpha.2016.07.005 Fabrication of an electrochemical sensor for determination of doxorubicin in human plasma and its interaction with DNA
10.1016/j.snb.2020.129120 Electrochemical detection of interaction between daunorubicin and DNA by hybrid nanoflowers modified graphite electrodes
10.1016/j.bioelechem.2014.06.002 Redox mechanism of anticancer drug idarubicin and in-situ evaluation of interaction with DNA using an electrochemical biosensor
10.1016/j.apsusc.2008.06.132 The amplification effect of functionalized gold nanoparticles on the binding of anticancer drug dacarbazine to DNA and DNA bases
10.1016/j.bpc.2005.04.009 Interaction of anticancer drug mitoxantrone with DNA analyzed by electrochemical and spectroscopic methods
We agree, and most of the cited papers were included, in order to emphasize the relevance of our workflow study. Our aim, were to use the carbon paste electrode (CPE) modified with calf thymus ds-DNA (CPDE) and computational methods to evaluate the drug-DNA interactions of different DNA target drugs and we also have included a non-anticancer, aiming to evaluate eventual correlations. There are not so many papers using electroanalysis and docking studies, and it is the first time that all these drugs are evaluated and compared in the same study.
Round 2
Reviewer 1 Report
Accept
Author Response
We appreciate the cooperation and acceptance
Reviewer 2 Report
1. ."DP voltammograms obtained for CPE (---) CPDE (---) in pH 7.0 0.1 M PBS after 60 seconds immersion in 1 mM 147
solutions of IDA, MIT, MTX, DAR , and CBP." here, which one is CPE which one is CPDE? please make different"CPE (---) CPDE (---)" in the related Figure caption.
Author Response
The Figures has been adjusted
Reviewer 3 Report
In this study, the author intends to evaluate the interaction between DNA and different drugs using electrochemical and docking method. In my opinion, the manuscript is not suitable for publication in Molecules Journal. However, I have specific comments
The novelty of this study is low because the interaction of these drugs has already been studied by various methods, especially the electrochemical method.
In this study, the oxidation signal of guanine, which appears in the range of 880 to 1000 mV, was not considered that can interfere with the signal of different drugs.
The procedure for fabrication of CPED is not clear.
The effect of the pH has not been investigated in this study.
Table 1 is quite confusing and does not provide useful information to the reader.
Author Response
The novelty of this study is low because the interaction of these drugs has already been studied by various methods, especially the electrochemical method.
Answer: The novelty of the study is a simpler and faster approach in the use of carbon paste electrode for the evaluation of drug-dsDNA intercalation, in this way we may use double-stranded DNA, which does not require denaturation procedures, thus facilitating the preparation and, consequently, the time of the tests. The goal is a rapid methodology that obtains results that corroborate theoretical methods, in this case molecular docking.
In this study, the oxidation signal of guanine, which appears in the range of 880 to 1000 mV, was not considered that can interfere with the signal of different drugs.
Answer: The interaction with adenine and guanine peaks was not evaluated because we use an intact double-stranded DNA, thus it is not possible to observe the signals from the bases of the DNA. The purpose of the manuscript is to demonstrate a new methodology with less preparation and trial time that corroborates with in silico methods, properly evaluating drug-DNA intercalation.
The procedure for fabrication of CPED is not clear.
Answer: It was rewritten, the preparation is simple, consisting in the addition of 50 microliters of 1% double-stranded DNA solution in the composition prepared in the carbon paste electrode.
The effect of the pH has not been investigated in this study.
Answer: A pH study was not evaluated as we adopted a range close to the physiological pH so that there were no changes in the DNA double helix structure.
Table 1 is quite confusing and does not provide useful information to the reader.
Answer: It was simplified, and the controversial statements removed